# Improving Separation Efficiency of Photogenerated Charges through Combination of Conductive Polythiophene for Selective Production of CH$_4$

Yiqiang Deng [1], Lingxiao Tu [2], Ping Wang [2], Shijian Chen [1], Man Zhang [2], Yong Xu [2,*] and Weili Dai [2,*]

[1] School of Chemical Engineering, Key Laboratory of Inferior Crude Oil Upgrade Processing of Guangdong Provincial Higher Education Institutes, Guangdong University of Petrochemical Technology, Maoming 525000, China

[2] Key Laboratory of Jiangxi Province for Persistent Pollutants Control and Resources Recycle, Nanchang Hangkong University, Nanchang 330063, China

* Correspondence: xu_yong001@163.com or xuyong01@nchu.edu.cn (Y.X.); wldai81@126.com or daiweili@nchu.edu.cn (W.D.)

**Abstract:** In today's society, mankind is confronted with two major problems: the energy crisis and the greenhouse effect. Artificial photosynthesis can use solar energy to convert the greenhouse gas CO$_2$ into high-value compounds, which is an ideal solution to alleviate the energy crisis and solve the problem of global warming. The combination of ZnO and polythiophenes (PTh) can make up for each other's drawbacks, thus improving the photoresponse behavior and separation efficiency of the photogenerated carriers. The PTh layer can transfer photogenerated electrons to ZnO, thereby extending the lifetime of the photogenerated charges. The production rate of CH$_4$ from the photoreduction of CO$_2$ with ZnO/PTh$_{10}$ is 4.3 times that of pure ZnO, and the selectivity of CH$_4$ is increased from 70.2% to 92.2%. The conductive PTh can absorb photons to induce $\pi$–$\pi^*$ transition, and the photogenerated electrons can transfer from the LUMO to the conduction band (CB) of ZnO, thus more electrons are involved in the reduction of CO$_2$.

**Keywords:** polythiophene; ZnO; CO$_2$ reduction; photocatalysis; CH$_4$

## 1. Introduction

"Learning from nature" is one of the main strategies for human social activities. In view of this, it is an ideal method to alleviate the current energy crisis and solve the greenhouse effect by simulating photosynthesis and taking sunlight as the driving force to convert the greenhouse gas CO$_2$ into high-value-added hydrocarbons [1,2]. Since the CO$_2$ molecule is thermodynamically stable with a bond energy of C=O up to 750 kJ mol$^{-1}$, few catalysts can directly reduce it by one electron [3,4]. Nevertheless, the photosynthesis of green plants is a multi-electron and multi-proton process, avoiding the high reduction potential for single electron reduction ($-1.9$ V vs. NHE) [5,6]. Therefore, photocatalysis is also an effective method for achieving CO$_2$ reduction. Predictably, the high-performance photoreduction of CO$_2$ is dependent on photocatalysts with efficient light adsorption, fast photogenerated carrier separation and the appropriate redox potential.

Zinc oxide (ZnO), with low toxicity and chemical stability, is widely used as a gas sensor, solar cell, field-effect transistor, piezoelectric generator, light-emitting diode (LED), photodetector, etc. [7]. However, the common drawback of semiconductors is the high recombination rate of the photogenerated charge carriers, leading to unconvincing photocatalytic performance [8]. On the other hand, the inherently wide bandgap (3.2 eV) permits activation only in the ultraviolet region, restricting the utilization of clean and abundant solar light [9,10]. In order to make up for the shortcomings of ZnO to suppress the recombination of photogenerated electrons and holes, researchers have taken many measures, such as doping, forming heterojunctions, loading noble metals, manufacturing

defects and so on [11–13]. However, the poor conductivity of ZnO limits the rapid transfer of photogenerated electrons and holes.

A conductive polymer with a spatially extended π-conjugated electron system has drawn more and more attention and may act as a promising alternative to traditional inorganic semiconductors [14]. The preparation of conducting polymers is simple and achieved through chemical or electrochemical methods. Consequently, conductive polymers have been applied in energy storage, sensors, environmental protection and other fields [15,16]. More importantly, conductive polymers have a small bandgap, which allows them to absorb visible light from the sun. Recently, the combination of conductive polymers and semiconductor materials has improved the utilization of photogenerated charges, promoting the degradation of dyes and water decomposition [17–19]. However, the research on conductive polymers in the field of photocatalysis is not deep enough, and some problems, such as the impact on product selectivity, are not clear.

Given the above analysis, combining the stable inorganic semiconductor ZnO with a flexible polymer with good conductivity to study the activity and product selectivity of photocatalytic $CO_2$ reduction may be a promising method. PTh are one class of conductive polymers, which show exceptional thermal and chemical stability and remarkable optical properties, with a bandgap of 2.0 eV [20]. Therefore, constructing a composite material by polymerizing a layer of thiophene on the surface of ZnO with a wide bandgap is conducive to accelerating the transfer of photogenerated electrons and improving the light absorption capacity. The formation of a large number of photogenerated electrons is beneficial for promoting the production of the 8-electron reduction product of $CH_4$. Finally, the production rate of $CH_4$ from the photoreduction of $CO_2$ with the composite material is 4.3 times that of pure ZnO, and the selectivity of $CH_4$ is increased from 70.2% to 92.2%.

## 2. Results and Discussion

### 2.1. Photocatalyst Characterization

First, porous ZnO nanosheets were prepared using a hydrothermal method [21], as shown in Figure 1a. Subsequently, the thiophene monomers were polymerized on the surface of ZnO in an ice water bath using ammonium persulfate as a polymerization agent to form the composite material of $ZnO/PTh_x$. A TEM image of the pure ZnO nanosheets is shown in Figure S1a, and the size distribution of the nanosheets ranges from 300 to 500 nm. The lattice fringe spacings of 0.260 and 0.247 nm in the HRTEM images (Figure S1b–d) correspond to the (002) and (101) crystal planes of ZnO, respectively. High-angle annular dark-field scanning transmission electron microscopy (HAADF-STEM) and the corresponding elemental mapping images are shown in Figure S1d–f, and the porous structure of ZnO and the distribution of the Zn and O elements can be clearly observed.

The XRD patterns of ZnO and $ZnO/PTh_{10}$ are displayed in Figure 1b, and ZnO matches well with the hexagonal crystal structure (JCPDS No. 70-2551). The high-intensity peaks indicate that the synthesized ZnO and $ZnO/PTh_{10}$ materials have good crystallinity. The crystallinity and characteristic peaks of the ZnO materials have not changed significantly before and after the composite of PTh, suggesting the introduction of the conductive polymer PTh will not change the crystal structure of ZnO. The slight decrease in the diffraction peak intensity of $ZnO/PTh_{10}$ may be due to the PTh on the surface masking part of the signals. The TEM image (Figure 1c) shows that there is no significant difference between the morphology of $ZnO/PTh_{10}$ and pure ZnO, both of which are nanosheets. The lattice spacing of 0.26 nm (Figure 1d) belongs to the (002) crystal plane of ZnO. In addition, a layer of amorphous material with a thickness of 4 nm can be observed on the surface of the composite material, which is assigned to a polymer of thiophene. Thus, the thin thickness of this polymer layer will not affect the catalytic ability of ZnO. The HAADF-STEM image of $ZnO/PTh_{10}$ (Figure 1e) indicates that the porous structure of the ZnO remains unchanged. Furthermore, it can be seen that the C and S elements contained in the PTh are evenly distributed on the surface of the ZnO by element mapping analysis of the $ZnO/PTh_{10}$ material (Figure 1f–i).

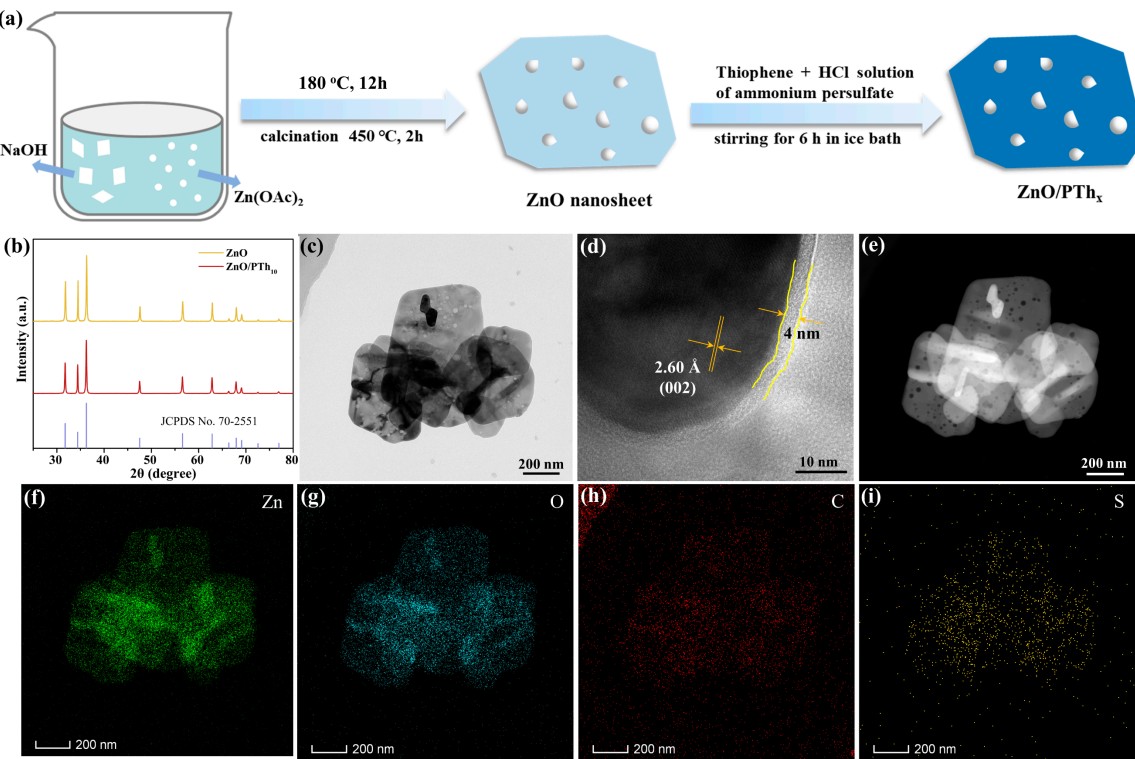

**Figure 1.** Schematic diagram of preparation of ZnO/PTh$_x$ (**a**). XRD patterns of ZnO and ZnO/PTh$_{10}$ (**b**). TEM image of ZnO/PTh$_{10}$ (**c**) and HRTEM image of ZnO/PTh$_{10}$ (**d**). HAADF-STEM of ZnO/PTh$_{10}$ (**e**) and corresponding elemental mapping images (**f–i**).

The infrared spectra of ZnO and ZnO/PTh$_{10}$ are represented in Figure S2. A broad peak between 3200 and 3700 cm$^{-1}$ can be observed, which is due to the stretching vibration of the hydroxyl group (–OH) generated by the coordination of water molecules with the ZnO sample [21]. The peak between 1500 and 1650 cm$^{-1}$ is attributed to the bending vibration of the –OH. A new peak appears near 1100 cm$^{-1}$ in the spectrum of ZnO/PTh$_{10}$, which is attributed to the vibration modes of the water molecules [21], indicating that ZnO/PTh$_{10}$ is more hydrophilic, and the PTh formed on the surface can be used as a mass transfer channel. The infrared peaks have not changed significantly after the introduction of PTh, which indicates that the core crystal structure of ZnO has not changed, which is consistent with the above XRD analysis result.

In order to study the pores and specific surface area of the prepared materials, N$_2$ adsorption–desorption tests were conducted. As shown in Figure 2a, both materials of ZnO and ZnO/PTh$_{10}$ possess hysteresis loops, and the isotherms belong to the IV-type [22], which suggests the as-prepared materials have abundant mesopores. The specific surface areas of the as-prepared samples are calculated according to the BET (Brunauer–Emmett–Teller) equation, and the corresponding results are 42 and 51 m$^2$ g$^{-1}$ for ZnO and ZnO/PTh$_{10}$, respectively. The PTh layer is a porous material, which is conducive to the adsorption of gas molecules. The pore size distributions of the as-prepared materials are calculated by the Barrett–Joyner–Halenda (BJH) method from the desorption branch of the isotherm curve, and the corresponding results are displayed in Figure 2b. The average pore sizes of ZnO and ZnO/PTh$_{10}$ are 15 and 17 nm, respectively. Therefore, the large specific surface area and porous structure are conducive to the transfer of mass and charges.

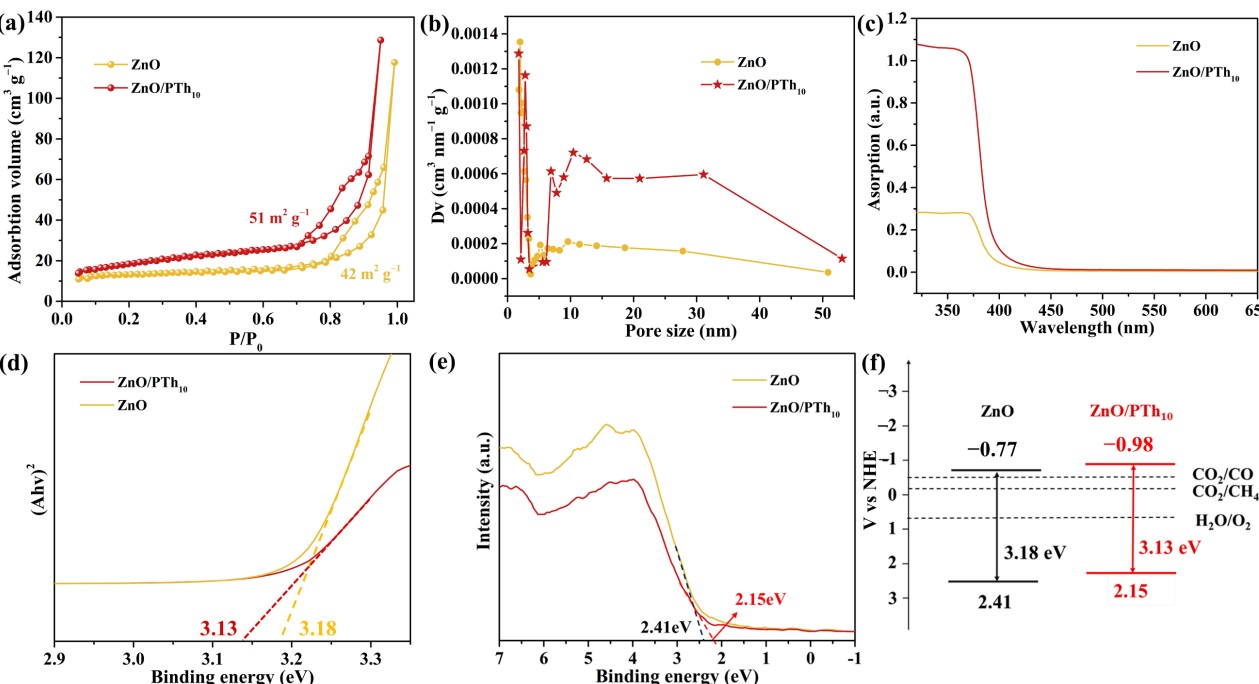

**Figure 2.** N$_2$ adsorption–desorption isotherm diagrams of ZnO and ZnO/PTh$_{10}$ (**a**) and corresponding pore size distribution (**b**). Ultraviolet–visible DRS of ZnO and ZnO/PTh$_{10}$ (**c**) and corresponding plots of ($\alpha$h$\upsilon$)$^2$ versus energy (h$\upsilon$) for the band gap energy (**d**). Valence band XPS spectra (**e**). Band structure alignments of ZnO and ZnO/PTh$_{10}$ (**f**).

Ultraviolet–visible spectroscopy (UV–vis) is commonly used to evaluate the light absorption performance of materials. As depicted in Figure 2c, ZnO/PTh$_{10}$ has a significantly stronger absorption capacity for photons than ZnO in the wavelength range of 380 to 500 nm. The redshift of composite materials is beneficial for the absorption of visible light, which is related to the excitation of PTh from the highest occupied molecular orbital (HOMO) to the lowest unoccupied molecular orbital (LUMO) [23]. The bandgaps were determined using the Tauc/Davis-Mott model with the equation ($\alpha$h$\upsilon$)$^{1/n}$ = A(h$\upsilon$ − E$_g$) [24]. The value of the exponent n denotes the nature of the material and is defined as 0.5 for a directly allowed transition like ZnO. The fitting result reveals that the bandgaps of ZnO and ZnO/PTh$_{10}$ are approximately 3.18 (close to the theoretical value of 3.20 eV) and 3.13 eV (Figure 2d), respectively. The reference values of the valence band (E$_{VB}$) were measured by XPS valence spectra (Figure 2e), and the obtained values are 2.41 and 2.15 eV for ZnO and ZnO/PTh$_{10}$, respectively. The conduction band values (E$_{CB}$) can be calculated by the difference between E$_{VB}$ and E$_g$, and the final results are displayed in Figure 2f. The conduction band potential values of ZnO and ZnO/PTh$_{10}$ are both lower than the theoretical potentials for reducing CO$_2$ to CO and CH$_4$, thus ZnO and ZnO/PTh$_{10}$ have the potential to convert CO$_2$ to CO and CH$_4$.

## 2.2. Efficient Separation of Photogenerated Charges

XPS was applied to further analyze the effect of PTh introduction on the valence state, chemical composition and surface chemical state of ZnO. There are two characteristic peaks of pure phase ZnO at 1021.1 eV (Zn 2p$_{3/2}$) and 1044.1 eV (Zn 2p$_{1/2}$) [25] in Figure 3a. After adding PTh, the fitting peaks of Zn 2p shift to the direction of small binding energy (1020.9 and 1043.9 eV), which implies that ZnO gains electrons from the PTh layer, resulting in a decrease in the valence state of Zn. The peak at 529.3 eV in the O 1s spectra is attributed to the binding energy of Zn–O in the characteristic hexagonal wurtzite [26–28]. In addition, the peak located at 531.5 eV is related to the bonding of the C=O of the adsorbed CO$_2$ molecules [26]. When the surface of ZnO is coated with PTh, the fitting peaks of O 1s

also shift to the direction of low binding energy, which further confirms that ZnO obtains electrons from the PTh layer. In addition, a new peak appeared at 533.4 eV for ZnO/PTh$_{10}$, indicating the presence of thiophene energy bonds. Hence, the XPS analysis results certify that the PTh layer can transfer some electrons to ZnO after the composite material is constructed, thus improving the electron density of the active sites in ZnO.

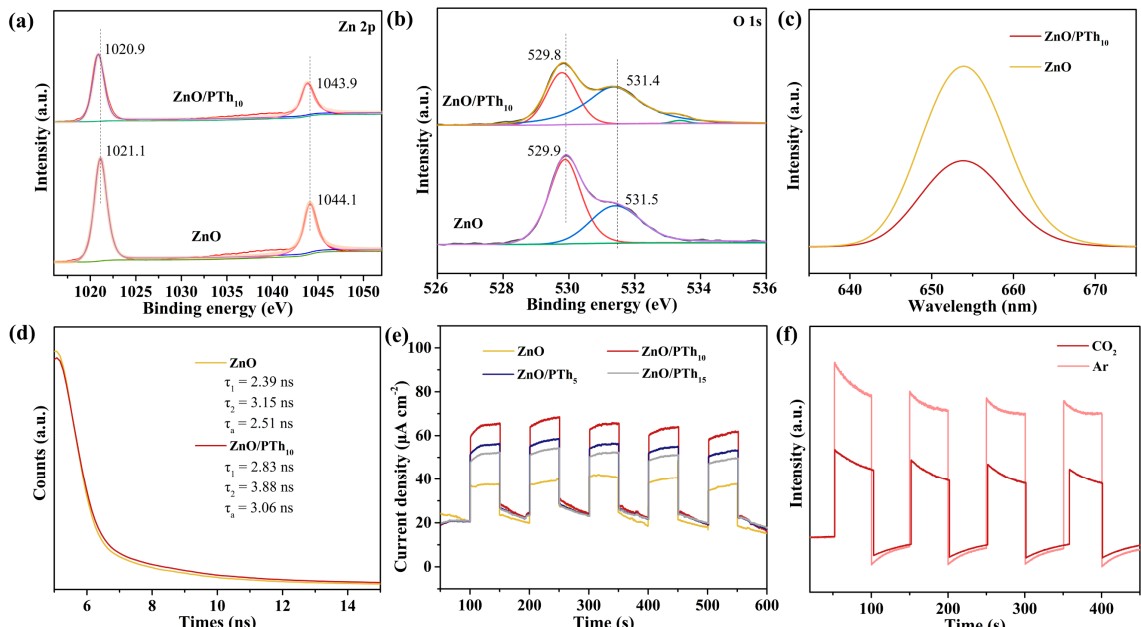

**Figure 3.** High-resolution XPS spectra of (**a**) Zn 2p and (**b**) O 1s. Steady-state PL spectra (**c**) and time-resolved PL decay spectra (**d**) for ZnO and ZnO/PTh$_{10}$. (**e**) Transient photocurrent spectra. (**f**) Transient photocurrent spectra with ZnO/PTh$_{10}$ as catalyst under the atmosphere of Ar and CO$_2$.

Steady-state photoluminescence (PL) spectroscopy is commonly used to investigate the separation efficiency of photogenerated charge carriers between the interfaces. The PL spectra of the ZnO and ZnO/PTh$_{10}$ materials are shown in Figure 3c. Both materials produce emission peaks with similar shapes at 654 nm, with the utilization of a 325 nm laser as the excitation light source. The emission peak intensity of the material containing the PTh layer is significantly reduced. In principle, the lower the recombination rate of the photogenerated electrons and holes, the lower the intensity of the PL emission peak [29,30]. As a result, the separation efficiency of the photogenerated charges in ZnO/PTh$_{10}$ greatly improves after being excited by light, and more electrons can participate in the CO$_2$ reduction reaction. The PL decay lifetime was employed to further explore the separation of photogenerated carriers. As presented in Figure 3d, the fitting results indicate that the PL decay lifetime of ZnO/PTh$_{10}$ is 3.06 ns, which is longer than that of pure ZnO (2.51 ns). The extension of the lifetime means that more photogenerated electrons transfer from the PTh layer to ZnO, which is conducive to improving the photoreaction rate as well as the product selectivity.

As shown in Figure 3e, the transient photocurrent response curves of ZnO and ZnO/PTh$_x$ (x = 5, 10, 15) were tested under visible light excitation. The four materials exhibit an instantaneous current response when intermittently switching the light five times. The produced photocurrent was mainly the result of photoinduced electrons diffusing to the FTO (fluorine-doped tin oxide) [31]. Therefore, the enhanced photocurrent implies that more effective charge transfer is achieved after covering the surface of ZnO with a layer of PTh. ZnO/PTh$_{10}$ exhibits the strongest light response signal; hence, ZnO/PTh$_{10}$ is the optimal photocatalyst. Electrochemical impedance spectroscopy (EIS) was further used to verify the above result, and the final result is consistent with the conclusion of the photocurrent (Figure S3). To probe the interfacial charge kinetics for the photoreduction

of $CO_2$, the transient photocurrent responses were measured in the electrolytes saturated with Ar and $CO_2$. As depicted in Figure 3f, the photocurrent in the $CO_2$ atmosphere is lower than that in the Ar atmosphere, indicating a decrease in the photocurrent results from the competitive electron transfer from $ZnO/PTh_{10}$ to the chemisorbed $CO_2$. The efficient electron delivery is advantageous to $CO_2$ activation and the reduction process [32].

### 2.3. Photocatalytic $CO_2$ Reduction Activity

The photocatalytic reduction of $CO_2$ was carried out in an aqueous solution with triethanolamine (TEOA) as the sacrificial agent under photoexcitation. First, by comparing the photocatalytic performance of a series of composite materials with different amounts of PTh, it can be further explained that the PTh layer is the key factor to improve the photocatalytic performance of $CO_2$ reduction. As displayed in Figure 4a, compared with pure ZnO, the photocatalysis activities of composite materials are improved in varying degrees after adding different proportions of PTh. In particular, the selectivity of $CH_4$ in the products is significantly increased, and $ZnO/PTh_{10}$ exhibits the highest catalytic activity, which is consistent with the results of the photogenerated charge separation efficiency. The difference in the photocatalytic performance of the composite materials may be due to the influence of the amount of PTh on the surface on the migration of photogenerated charges and the transfer of reactant molecules. And a suitable amount of coating can maximize the photocatalytic performance without affecting the light absorption ability.

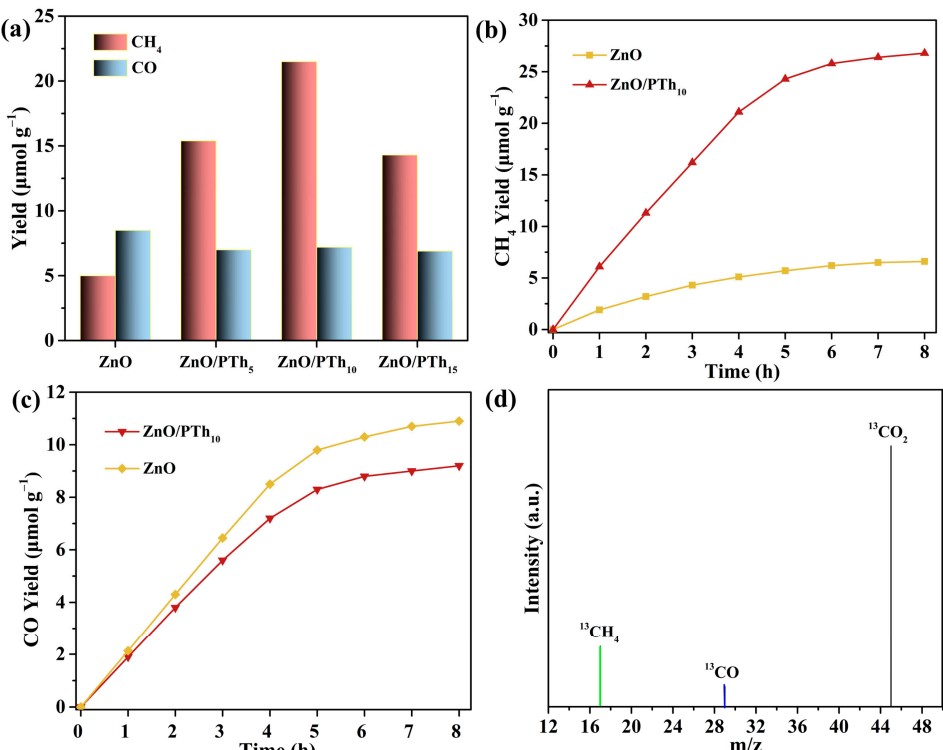

**Figure 4.** The effect of different PTh loadings on the photocatalytic activity of ZnO (**a**); the illumination time is 4 h. The amount of $CH_4$ (**b**) and CO (**c**) produced by photocatalytic reduction of $CO_2$ over time. (**d**) Mass spectrometry analysis of reduction products.

The curves of the amount of reduction products over time are shown in Figure 4b,c. In the first 4 h, the rate of $CH_4$ production with ZnO as a catalyst gradually decreases, while $ZnO/PTh_{10}$ shows no obvious change, which indicates that the introduction of PTh can reduce photocorrosion and improve stability. The production rate of $CH_4$ increases from 5.0 μmol $g^{-1}$ to 21.4 μmol $g^{-1}$ in the first 4 h, and the activity of $CH_4$ production over the composite material of $ZnO/PTh_{10}$ is 4.3 times higher than that of ZnO. However, the rate of CO production over $ZnO/PTh_{10}$ decreases instead, which means that the concentration

of photogenerated electrons increases after covering the PTh layer, tending to form a multi-electron reduction product ($CH_4$) [33]. According to the amount of electrons transferred, the selectivity of $CH_4$ increases from 70.2% to 92.2%. After 5 h of illumination, the number of products increased very little over time. The possible reasons for the decrease in products are a drop in $CO_2$ concentration in the quartz tube and the continuous consumption of the sacrificial agent of TEOA. The following scientific controls were carried out to verify that ZnO/PTh$_{10}$ was a true photocatalyst. As shown in Figure S4, a photocatalyst, $CO_2$ and light are essential conditions for achieving photocatalytic $CO_2$ reduction. In addition, an isotopic labeling experiment confirmed that the carbon in the reduction products is derived from $CO_2$ (Figure 4d). There are three obvious signal peaks at the mass-to-charge ratios ($m/z$) of 17, 29 and 45, which correspond to $^{13}CH_4$, $^{13}CO$ and $^{13}CO_2$, respectively. The cyclic experimental result of ZnO/PTh$_{10}$ is shown in Figure S5, and the photocatalytic performance did not significantly decrease after four cycles, further proving the high stability of the composite material. In addition, the characterizations of ZnO/PTh$_{10}$ after the cyclic experiment are shown in Figure S6. As can be seen, the position and shape of the diffraction peaks after cycling remain unchanged, and the morphology and lattice of ZnO/PTh$_{10}$ are also consistent with those of the unused material. The above results indicate that ZnO/PTh$_{10}$ has excellent stability.

### 2.4. Photocatalytic Mechanism

The wide bandgap of ZnO is weak in the absorption of visible light, while PTh can generate carriers when excited by light, but the recombination rate of carriers is fast [34]. The combination of ZnO and PTh can make up for each other's shortcomings and improve the photoresponse behavior. The flat-band potentials of ZnO and ZnO/PTh$_{10}$ were measured using the Mott–Schottky method (Figure 5a). The obtained potentials of $-0.75$ V and $-0.85$ V (vs. NHE, pH 7.0) for ZnO and ZnO/PTh$_{10}$, respectively, indicate that the two photocatalysts possess suitable redox potential to catalyze the reduction of $CO_2$ to CO (E = $-0.53$ V) and $CH_4$ (E = $-0.24$ V) [35]. The possible photocatalytic mechanism is speculated, as shown in Figure 5b. Conductive PTh can absorb photons to induce π–π* transition, and the photoexcited electrons generated by PTh are transferred from LUMO to the CB of ZnO and participate in the reduction of $CO_2$ [19]. The holes in the VB of ZnO are transferred to the HOMO of PTh, thus improving the separation efficiency of the photogenerated electrons and holes. Ultimately, more photogenerated electrons take part in the reduction reaction, boosting the reduction rate of $CO_2$ and the selectivity of $CH_4$.

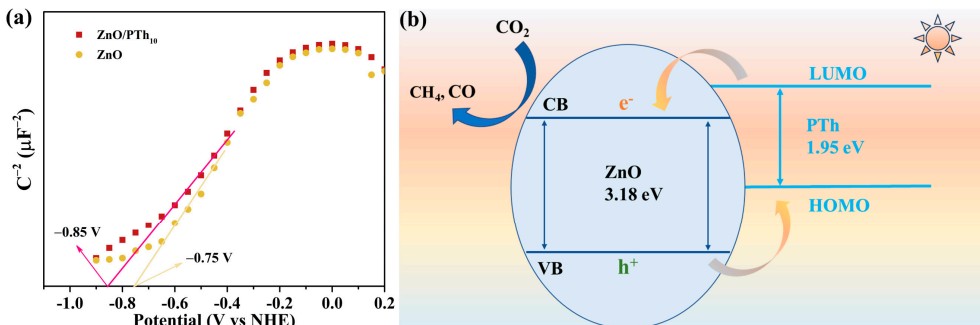

**Figure 5.** (**a**) Mott–Schottky plots of ZnO and ZnO/PTh$_{10}$. (**b**) Speculated mechanism for photocatalytic reduction of $CO_2$ over ZnO/PTh$_{10}$ under visible light irradiation.

## 3. Materials and Methods

### 3.1. Material Preparation

#### 3.1.1. Synthesis of ZnO

In a typical synthesis, 3.45 g of Zn(OAc)$_2$ (AR) and 2.5 g of NaOH (AR) were dissolved in 80 mL of deionized water under stirring for 2 h. The obtained mixture was transferred to a 100 mL stainless autoclave with a heating rate of 5 °C min$^{-1}$ and kept at 180 °C for 12 h.

After natural cooling to room temperature, the mixture was centrifuged, and the resulting solid was washed several times with acetone (AR) and dried. Finally, the obtained sample was calcined at 450 °C for 2 h with a heating rate of 5 °C min$^{-1}$ in an Ar atmosphere.

### 3.1.2. Preparation of ZnO/PTh$_x$

A total of 1.0 g of ZnO material was uniformly dispersed in 50 mL of ethanol, and a certain amount of the monomer thiophene was added under stirring at 0 °C (mass ratios of 1:5, 1:10 and 1:15, respectively). After half an hour, 2 mL of hydrochloric acid (2 mol L$^{-1}$) solution containing 0.135 g of ammonium persulfate (pH = 2) was added to the above mixture. The above mixture was continuously stirred under ice bath conditions and washed three times alternately with deionized water and ethanol (120 mL every time) after stirring for 6 h. The obtained solid was dried in a vacuum oven, which was labeled as ZnO/PTh$_x$ (X = 5, 10, 15).

### 3.2. *Characterization*

The crystallinity of the samples was obtained using a Bruker AXS-D8 Advance X-ray diffractometer (XRD) (Bruker, Billerica, MA, USA) with Cu K$\alpha$ radiation in the 2θ range of 5–80°. Transmission electron microscopy (TEM) and high-resolution TEM (HRTEM) images were recorded using a JEOL 2100 microscope (Hitachi, Tokyo, Japan) operated at a 200 kV accelerating voltage. Nitrogen adsorption–desorption isotherms were collected using a NOVA1000 nitrogen adsorption apparatus (Quantachrome, Boynton Beach, FL, USA) at 77 K. The conditions for the desorption experiment were a temperature of 120 °C, a pressure of 0.1 MPa, and a time of 6 h. The pore size distributions of the as-prepared materials were calculated using the Barrett–Joyner–Halenda (BJH) method from the desorption branch of the isotherm curve. X-ray photoelectron spectroscopy (XPS) was performed using a Kratos Axis Ultra DLD spectrometer (Shimadzu, Kyoto, Japan) with a monochromatic Al K$\alpha$ X-ray source. The ultraviolet–visible diffuse reflectance spectra (DRS) were recorded on a spectrophotometer (Hitachi UV-3010, Tokyo, Japan) using BaSO$_4$ as a reference. Steady-state and time-resolved photoluminescence (PL) spectroscopy (Edinburgh FS5, Livingston, UK) was used to obtain the transient spectra of the specimen at room temperature.

Electrochemical measurements were performed with a CHI 660D electrochemical workstation (Chenhua, Shanghai, China) in a standard three-electrode system. FTO deposited with the as-prepared sample served as a working electrode with an active area of 1.0 cm$^2$, while the counter and the reference electrodes were graphite and a saturated calomel electrode, respectively. Time-dependent photocurrent measurements were performed using a 300 W Xe lamp equipped with an ultraviolet cut-off filter ($\lambda \geq 420$ nm). The Mott–Schottky curves were recorded at a fixed frequency of 1000 Hz with a 5 mV amplitude. The electrochemical impedance spectroscopies (EIS) were recorded in the frequency range from 10 kHz to 0.01 Hz, and the applied bias voltage and ac amplitude were set at open-circuit voltage. A 0.5 M Na$_2$SO$_4$ aqueous solution was used as the electrolyte.

### 3.3. *Photocatalytic Tests*

The photocatalytic CO$_2$ reduction reaction was carried out in a 55 mL quartz tube, and the products were tested using a gas chromatograph (Zhejiang Fuli Analytical Instrument Co., Ltd., Taizhou, China) with TCD and FID detectors. First, 10 mg of the photocatalyst, 10 mL of deionized water and 1 mL of triethanolamine (TEOA) were added to the quartz tube. Subsequently, the quartz tube was subjected to ultrasonic treatment for 15 min to evenly disperse the material. Then, high-purity CO$_2$ (99.999%) was bubbled into the quartz tube (30 mL min$^{-1}$), and, after 30 min, the quartz tube was completely sealed using a silicone plug and sealing film. After 30 min of bubbling, the pH of the reaction mixture decreased from 10.50 to 7.95. The sealed quartz tube was illuminated with white LED lamps with a power of 200 W. During this period, the products in the quartz tube were detected and analyzed every 1 h.

$^{13}CO_2$ was used to replace the reactant of $CO_2$, while the other reaction conditions remained unchanged. After 4 h of illumination, a gas chromatography-mass spectrometry device was used to separate and detect the final products.

To evaluate the stability of the catalyst, the partition strategy was employed to compensate for the filtration and transferred loss of $ZnO/PTh_{10}$ among the cycles. In each cycle, 10 mg of $ZnO/PTh_{10}$ was taken out from a total of 60 mg for the photoreaction. Then, after each 10 mg cycle, a parallel experiment was carried out for the rest of the catalysts under the same conditions. Subsequently, all the catalysts were collected together after washing and drying, and another 10 mg of $ZnO/PTh_{10}$ was taken out for the following run.

## 4. Conclusions

A conductive polymer with a spatially extended π-conjugated electron system is beneficial for the rapid transfer of electrons. The combination of ZnO and PTh can compensate for the shortcomings of a broad bandgap and the easy recombination of photogenerated carriers. The experimental results of the photocurrent and EIS show that the separation efficiency of the photogenerated charge is the highest when the thickness of the PTh layer is 4 nm ($ZnO/PTh_{10}$). An increase in the specific surface area and average pore size of $ZnO/PTh_{10}$ is conducive to the transfer of mass and charges. The PTh layer can transfer electrons to ZnO by XPS analysis, thus the lifetime of the photogenerated charge is lengthened, which is beneficial to boost the photocatalytic activity. An analysis of the band structure and flat-band potentials indicated that $ZnO/PTh_{10}$ can reduce $CO_2$ to CO and $CH_4$. Finally, the production rate and selectivity of $CH_4$ greatly improved.

**Supplementary Materials:** The following supporting information can be downloaded at https://www.mdpi.com/article/10.3390/catal13071142/s1, Figure S1: TEM images of ZnO; Figure S2: Infrared spectra of ZnO and $ZnO/PTh_{10}$; Figure S3: Nyquist plots of ZnO and $ZnO/PTh_x$; Figure S4: Effect of different reaction conditions on the photocatalytic activity of $ZnO/PTh_{10}$; Figure S5: Cycle test for $ZnO/PTh_{10}$. Figure S6. (a) XRD patterns and (b) TEM images of $ZnO/PTh_{10}$ after cyclic experiment.

**Author Contributions:** Investigation, experimental work, analysis and discussion, data curation, writing, Y.D.; methodology, software, formal analysis, L.T.; characterization, analysis and discussion, P.W.; analysis and discussion, S.C.; data curation, M.Z.; writing—review and editing, funding acquisition, Y.X.; supervision, writing—review and editing, funding acquisition, project administration, W.D. All authors have read and agreed to the published version of the manuscript.

**Funding:** This research was financially supported by the Guangdong Provincial Science and Technology Innovation Strategy Project (Grant No. 2022DZXHT021), the Maoming Science and Technology Special Project (Grant No. 2022S049), the 2022 "Sail Plan" Project of Maoming Green Chemical Industry Research Institute (Grant No. MMGCIRI-2022YFJH-Z-001), the Jiangxi Provincial Natural Science Foundation (Grant Nos. 20224BAB214026 and 20224ACB203001) and the Key Research and Development Program of Jiangxi Province (Grant No. 20192ACB70009).

**Data Availability Statement:** The data presented in this study are available upon request from the corresponding author.

**Acknowledgments:** The authors are grateful for the financial support received for the project.

**Conflicts of Interest:** The authors declare no conflict of interest.

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
