# Peer review of "Improving Separation Efficiency of Photogenerated Charges through Combination of Conductive Polythiophene for Selective Production of CH4"

_catalysts, doi:10.3390/catal13071142_

Round 1
Reviewer 1 Report
This study “Improving Separation Efficiency of Photogenerated Charges 2 through Combination of Conductive Polythiophene for Selective Production of CH4” presents synthesis, characterisation and application of Zinc oxide combined with polythiophenes, ZnO/PTh, in photoreduction of CO2 to selectively reduce CH4. A thorough characterisation of the materials was performed by TEM, XRD, IR, N2 adsorption, ultravioleta, XPS, photocurrent. The photocatalytic activity for CO2 reduction was measured with different solids, conditions and cycles. This is an interesting work but a major revision must be performed to include significant information before considering its publication. The following comments should be addressed (please include in your answers the new text added in the main manuscript, if needed):
Major points are:
The reaction described in Figures 4b and 4c is clearly not finished. Results of a longer reaction should be included in this manuscript in order to observe reaction equilibrium or depletion.
Include in line 197 an explanation for the compound providing the H atoms to reduce CO2 to CH4 and the compound getting the O from CO2. The concentration evolution of these compounds should be followed with time as well as CO2 concentration.
Inlcude ZnO/PTh10 characterisation after reaction to study its stability as catalyst-
Minor points are:
1.- English should be reviewed
“Since CO2 molecule is thermodynamic stability” Since CO2 molecule is thermodynamically stable
“the common drawback of semiconductors is that the high recombination rate of photogenerated charge carriers” the common drawback of semiconductors is the high 39 recombination rate of photogenerated charge carriers
“Given the above analysis, combining stable inorganic semiconductor ZnO with flexible polymer with good conductivity to study the activity and product selectivity of photocatalytic CO2 reduction.” This sentence misses a consequence.
“Therefore, constructing composite material by polymerizing a layer of thiophene on the surface of ZnO with wide bandgap, which is conducive to accelerating the transfer of photogenerated electrons and improving light absorption capacity. This sentence misses a consequence.
“As depicted in Figure 3f, the photocurrent in CO2 atmosphere is lower than that in Ar atmosphere, manifesting that the decrease in photocurrent results from the competitive electron transfer from ZnO/PTh10 to the chemisorbed CO2.” Review.
“The wide bandgap of ZnO is weak in absorption of visible light, while PTh can generate carriers when excited by light, but the recombination rate of carriers is fast” Review
Line 119. “The special surface area” The specific surface area
“The sealed quartz tube was illustrated with white LED lamps with the power of 200 W.” The sealed quartz tube was illuminated with white LED lamps with the power of 200 W.
2.- Explain in the text “ZnO's own shortcomings to suppress the recombination”
3.- Why ZnO/PTh10 isotherm does not reach the same final relative pressure as ZnO?
4.- Why ZnO/PTh10 adsorbs more N2 than ZnO? Include explanation in the text
5.- Line 181, include in the text what is it FTO?
7.- Include formula to calculate CO2 conversion, CH4 and CO yields, CH4 selectivity
8.- What is the time of reaction for results presented in Figure 4a? Include it in the text
9.- Include in the text the conditions applied to obtain the results of figure 4d
10.- Include in the text the conditions applied to obtain the results of figure S4.
11.- Line 251. For autoclave include heating rate and cool down method.
12.- Line 253 Include heating rate and atmosphere of calcination.
13.- Line 257. Include HCl concentration
14.- Line 267 Include degassing information, temperature, pressure, time.
15.- How do you calculate average pore sizes? Include explanation in the text
16.- Photocatalytic tests
Include how you calculated the amount of CH4 and CO from TCD and FID.
Include measures of pH after 30 minutes of CO2 bubbling
Include the flow of CO2 in the quartz tube
Include description of a cycle.
1.- English should be reviewed
“Since CO2 molecule is thermodynamic stability” Since CO2 molecule is thermodynamically stable
“the common drawback of semiconductors is that the high recombination rate of photogenerated charge carriers” the common drawback of semiconductors is the high 39 recombination rate of photogenerated charge carriers
“Given the above analysis, combining stable inorganic semiconductor ZnO with flexible polymer with good conductivity to study the activity and product selectivity of photocatalytic CO2 reduction.” This sentence misses a consequence.
“Therefore, constructing composite material by polymerizing a layer of thiophene on the surface of ZnO with wide bandgap, which is conducive to accelerating the transfer of photogenerated electrons and improving light absorption capacity. This sentence misses a consequence.
“As depicted in Figure 3f, the photocurrent in CO2 atmosphere is lower than that in Ar atmosphere, manifesting that the decrease in photocurrent results from the competitive electron transfer from ZnO/PTh10 to the chemisorbed CO2.” Review.
“The wide bandgap of ZnO is weak in absorption of visible light, while PTh can generate carriers when excited by light, but the recombination rate of carriers is fast” Review
Line 119. “The special surface area” The specific surface area
“The sealed quartz tube was illustrated with white LED lamps with the power of 200 W.” The sealed quartz tube was illuminated with white LED lamps with the power of 200 W.
Author Response
Manuscript ID: catalysts-2480756
Dear Editor and Reviewers,
Thanks so much for your efforts on our paper entitled “Improving Separation Efficiency of Photogenerated Charges through Combination of Conductive Polythiophene for Selective Production of CH4”. We would like to thank all the reviewers sincerely for their time and efforts on our manuscript. All the comments are constructive and helpful for improving the quality and readability of our manuscript. We have carefully revised the article and answered the questions according to the reviewers’ comments, and all the changes are highlighted in yellow in the revised version. Our responses to the reviewers’ comments point by point are appended below.
Thank you and looking forward to hearing from you soon.
Best regards!
Yours sincerely,
Yong Xu
Listed below are our responses:
Reviewer #1
Major points are:
Comment: The reaction described in Figures 4b and 4c is clearly not finished. Results of a longer reaction should be included in this manuscript in order to observe reaction equilibrium or depletion. Include in line 197 an explanation for the compound providing the H atoms to reduce CO2 to CH4 and the compound getting the O from CO2. The concentration evolution of these compounds should be followed with time as well as CO2 concentration.
Response to comment: In general, the research on photocatalytic CO2 reduction focuses on the first 4 hours of the reaction (ref. 2, 5, 19 and 30). In the revised manuscript, the photocatalytic reaction time was extended to 8 hours, and the production rate of CH4 and CO significantly decreased after the 5th hour. After that, the amount of products increased very little over time. The possible reasons for the decrease in products are the drop of CO2 concentration in the quartz tube, and the continuous consumption of sacrificial agent of TEOA.
Comment: Inlcude ZnO/PTh10 characterisation after reaction to study its stability as catalyst-
Response to comment: The characterizations of ZnO/PTh10 after cyclic experiment are shown in Figure S6. As can be seen, the position and shape of the diffraction peaks after cycling remain unchanged. And the morphology and lattice of ZnO/PTh10 are also consistent with those of the unused material. The above results indicate that ZnO/PTh10 has excellent stability.
Figure S6. (a) XRD patterns and (b) TEM images of ZnO/PTh10 after cyclic experiment.
Minor points are:
Comment 1: English should be reviewed
(1)“Since CO2 molecule is thermodynamic stability” Since CO2 molecule is thermodynamically stable
(2)“the common drawback of semiconductors is that the high recombination rate of photogenerated charge carriers” the common drawback of semiconductors is the high recombination rate of photogenerated charge carriers
(3)“Given the above analysis, combining stable inorganic semiconductor ZnO with flexible polymer with good conductivity to study the activity and product selectivity of photocatalytic CO2 reduction.” This sentence misses a consequence.
(4)“Therefore, constructing composite material by polymerizing a layer of thiophene on the surface of ZnO with wide bandgap, which is conducive to accelerating the transfer of photogenerated electrons and improving light absorption capacity. This sentence misses a consequence.
(5)“As depicted in Figure 3f, the photocurrent in CO2 atmosphere is lower than that in Ar atmosphere, manifesting that the decrease in photocurrent results from the competitive electron transfer from ZnO/PTh10 to the chemisorbed CO2.” Review.
(6)“The wide bandgap of ZnO is weak in absorption of visible light, while PTh can generate carriers when excited by light, but the recombination rate of carriers is fast” Review
(7)Line 119. “The special surface area” The specific surface area
(8)The sealed quartz tube was illustrated with white LED lamps with the power of 200 W.” The sealed quartz tube was illuminated with white LED lamps with the power of 200 W.
Response to comment 1: Thanks for the valuable and constructive suggestion. Questions (1), (2), (7) and (8) have been modified in the revised manuscript, and the responses to other questions are as follows:
(3) Given the above analysis, combining stable inorganic semiconductor ZnO with flexible polymer with good conductivity to study the activity and product selectivity of photocatalytic CO2 reduction may be a promising method.
(4) Therefore, constructing composite material by polymerizing a layer of thiophene on the surface of ZnO with wide bandgap is conducive to accelerating the transfer of photogenerated electrons and improving light absorption capacity.
(5) Due to the strong ability of ZnO/PTh10 to activate CO2, some of the photogenerated charge carriers generated after photoexcitation will be used to catalyze the reduction of CO2 molecules, while the other part will participate in the electrochemical closed-loop circuit, resulting in a decrease in photocurrent intensity. This result is consistent with literature report (ref. 32).
(6) The bandgap of ZnO is 3.2 eV, which has weak absorption of visible light and strong absorption of ultraviolet light. PTh can generate carriers when excited by light (ref. 14, 17-19), however, the fast recombination rate of photogenerated electrons and holes leads to low photocatalytic activity.
The above issues have been modified accordingly in the revised manuscript.
Comment 2: Explain in the text “ZnO's own shortcomings to suppress the recombination”
Response to comment 2: The expression here is to inhibit the recombination of photogenerated charges in ZnO bulk phase, thereby promoting the participation of photogenerated charges in catalytic reactions and improving the utilization of light energy. Corresponding modifications have been made in the revised manuscript.
Comment 3: Why ZnO/PTh10 isotherm does not reach the same final relative pressure as ZnO?
Response to comment 3: The isothermal adsorption-desorption curves of ZnO/PTh10 was retested, and the results are shown in Figure 2a in the revised manuscript. The specific surface area of ZnO/PTh10 is 51 m2 g-1.
Comment 4: Why ZnO/PTh10 adsorbs more N2 than ZnO? Include explanation in the text
Response to comment 4: The polythiophene layer is porous material (it can be seen from the pore size distribution, Figure 2b), which is conducive to the adsorption of gas molecules. When PTh combined with ZnO, the porous structure enhances the N2 adsorption capacity. Corresponding modifications have been made in the revised manuscript.
Comment 5: Line 181, include in the text what is it FTO?
Response to comment 5: FTO is the abbreviation for fluorine-doped tin oxide, and is often used as conductive glass.
Question 6 does not exist or the number is incorrect.
Comment 7: Include formula to calculate CO2 conversion, CH4 and CO yields, CH4 selectivity
Response to comment 7: The products CH4 and CO were detected and quantitatively analyzed by using gas chromatography (FULI GC9790II). The calculation of product quantity was based on the peak area measured by gas chromatography using a calibration curve. The calibration curves were measured by quantitatively measuring the standard gases of CH4 and CO. And the obtained curves are shown in the following figure.
x is the micromolar amounts of the measured gases, and y is the peak area. The yields of CH4 and CO can be determined based on the formulas in the figure above and the measured peak areas. The peak areas measured in the experiment were all within the range of calibration curves. The selectivity of CH4 was calculated according to the amount of electron transfer: selectivity of CH4 (%) = 8n(CH4)]/[2n(CO) + 8n(CH4)] × 100%.
Comment 8: What is the time of reaction for results presented in Figure 4a? Include it in the text
Response to comment 8: The time for photocatalytic reaction is the first 4 hours. Corresponding modifications have been made in the revised manuscript.
Comment 9: Include in the text the conditions applied to obtain the results of figure 4d.
Response to comment 9: The 13CO2 was used to replace reactant of CO2, while other reaction conditions remained unchanged. After 4 hours of illumination, a gas chromatography-mass spectrometry device was used to separate and detect the final products. The content has been added to the section of Materials and Methods.
Comment 10: Include in the text the conditions applied to obtain the results of figure S4.
Response to comment 10: The experimental conditions and operations for reaction 1 are the same as the section of 3.3. Reaction 2 uses Ar gas instead of CO2, reaction 3 does not add photocatalysts, and reaction 4 does not undergo illumination. The contents have been added to the supporting information.
Comment 11: Line 251. For autoclave include heating rate and cool down method.
Response to comment 11: The obtained mixture was transferred into a 100 mL of stainless autoclave with a heating rate of 5 ℃/min and kept at 180 oC for 12 h. After natural cooling to room temperature, the mixture was centrifuged, and the resulting solid was washed several times with acetone and dried.
Comment 12: Line 253 Include heating rate and atmosphere of calcination.
Response to comment 12: Finally, the obtained sample was calcined at 450 °C for 2 h with a heating rate of 5 oC min-1 in Ar atmosphere.
Comment 13: Line 257. Include HCl concentration
Response to comment 13: The concentration of HCl is 2 mol L-1, and this content has been modified accordingly in the revised manuscript.
Comment 14: Line 267 Include degassing information, temperature, pressure, time.
Response to comment 14: Nitrogen adsorption-desorption isotherms were collected using a NOVA1000 nitrogen adsorption apparatus at 77 K. The conditions for the desorption experiment are temperature of 120 oC, pressure of 0.1 MPa, and time of 6 h.
Comment 15: How do you calculate average pore sizes? Include explanation in the text
Response to comment 15: The pore size distributions of the as-prepared materials were calculated by the Barrett–Joyner–Halenda (BJH) method from the desorption branch of the isotherm curve. Corresponding modifications have been made in the revised manuscript.
Comment 16: Photocatalytic tests
(1)Include how you calculated the amount of CH4 and CO from TCD and FID.
(2)Include measures of pH after 30 minutes of CO2 bubbling
(3)Include the flow of CO2 in the quartz tube
(4)Include description of a cycle.
Response to comment 16: (1) The amounts of CH4 and CO were calculated by calibration curves as answered in comment 7.
(2) After 30 min of bubbling, the pH of the reaction mixture decreased from 10.50 to 7.95, indicating that a large amount of CO2 was dissolved in the reaction solution.
(3) The flow rate of CO2 was 30 mL min-1. Corresponding modification has been made in the revised manuscript.
(4) To evaluate the stability of the catalyst, the partition strategy was employed to compensate for filtration and transferred loss of ZnO/PTh10 among the cycles. In each cycle, 10 mg of ZnO/PTh10 was taken out from a total of 60 mg for photoreaction, then after each 10 mg cycle, the parallel experiment was carried out for the rest of catalyst under the same conditions. Subsequently, all the catalysts were collected together, after washing and drying, another 10 mg of ZnO/PTh10 was taken out for the following run. Corresponding modifications have been made in the revised manuscript.

Reviewer 2 Report
With pleasure, I have reviewed the manuscript of the Article entitled “Improving separation efficiency of photogenerated charges through combination of conductive polythiophene for selective production of CH4”.
The paper deals with the synthesis, characterization and tests of ZnO-based photocatalysts for the reduction of CO2 into CO and CH4. The photocatalytic activity and selectivity were improved by adding polythiophenes to the ZnO. In my opinion, the novelty of the work is related to the insights provided on the use of conductive polymers in the photocatalytic reduction of CO2 to added-value products.
In conclusion, I recommend the article for publication in Catalysts after Major Revisions according to the comments in the attached file.

I warmly recommend a careful revision of the English grammar and of the sentences in the whole manuscript.
Author Response
Manuscript ID: catalysts-2480756
Dear Editor and Reviewers,
Thanks so much for your efforts on our paper entitled “Improving Separation Efficiency of Photogenerated Charges through Combination of Conductive Polythiophene for Selective Production of CH4”. We would like to thank all the reviewers sincerely for their time and efforts on our manuscript. All the comments are constructive and helpful for improving the quality and readability of our manuscript. We have carefully revised the article and answered the questions according to the reviewers’ comments, and all the changes are highlighted in yellow in the revised version. Our responses to the reviewers’ comments point by point are appended below.
Thank you and looking forward to hearing from you soon.
Best regards!
Yours sincerely,
Yong Xu
Listed below are our responses:
Reviewer #2
Comment 1: P1, L30: Please, pay attention to adverbs and adjectives. The sentence “Since CO2 molecule is thermodynamic stability with bond energy …” should be revised for instance as follows: “Since CO2 is a thermodynamically stable molecule …”.
Response to comment 1: Thanks for the valuable suggestion. This sentence has been modified in the revised manuscript.
Comment 2: P1, L30: I suggest avoiding the genitive for objects/things. So please, rewrite the sentence removing “ZnO’s”.
Response to comment 2: “ZnO’s own shortcomings” has changed to “shortcomings of ZnO”.
Comment 3: P2, L60-61: Please change “exceptionally” with “exceptional”. Please, pay attention to the use of adverbs and adjectives.
Response to comment 3: Thanks for the careful and precise reviewing. The adverb of “exceptionally” has changed to “exceptional”.
Comment 4: P2, L62-67: Please, do not summarise the results/conclusions with numbers. The last paragraph of the Introduction should be devoted to the aim of the research work, that the authors did not state clearly. Please, add the scope of the research work presented in the article.
Response to comment 4: The main purpose of this manuscript is to study the improvement of photocatalytic CO2 reduction activity and product selectivity. The combination of polythiophene and ZnO improves the separation efficiency of photogenerated charge, thus improving the photocatalytic activity and CH4 selectivity. These contents have been explained in the fourth paragraph. We also provided further explanations about the research work in the revised manuscript.
Comment 5: P7, L248-253: Which is the purity of each reactant? Which was the criteria for washing with acetone? How much acetone did you use? Moreover, which heating rate was used to calcine the sample?
Response to comment 5: Zn(OAc)2, NaOH and acetone are commercial chemicals with analytical grade purity. The obtained ZnO was washed three times with acetone, each time with 20 mL. The obtained sample of ZnO was calcined at 450 °C for 2 h with a heating rate of 5 oC min-1 in Ar atmosphere.
Comment 6: P7, L257: Please revise the sentence avoiding “with”.
Response to comment 6: Thanks for the careful and precise reviewing. The word of “with” has been removed.
Comment 7: P7, L259: Which was the criteria for washing with water and ethanol? Volume, concentration, pH?
Response to comment 7: The criteria for washing is the volume of the washing solution (120 mL every time). The above mixture was continuously stirred under ice bath conditions, and washed three times alternately with deionized water and ethanol (120 mL every time) after stirring 6 h. Corresponding modifications have been made in the revised manuscript.
Comment 8: P2, L75-79: In Figure S1 of the supplementary, the authors should identify each kind of planes in Figure S1b-c that are visible. Moreover, the markers are not clearly visible; please enlarge them.
Response to comment 8: The TEM images of Figure S1bc have been partially enlarged, as shown in the following figure b-d. And corresponding modification have been made in the revised manuscript.
Figure S1
Comment 9: P4, L116: Figure 2 should be introduced in the text before the images.
Response to comment 9: Figure 2 has been placed after its related contents.
Comment 10: P4, L125: I suggest approximating the average pore size to 15 nm and 17 nm as the uncertainty on the measure is larger than the second decimal unit.
Response to comment 10: Thanks for the careful and precise reviewing. The average pore sizes are only kept to integers.
Comment 11: P5, L185: Why ZnO/PTh10 showed the best performance? Which are the phenomena that reduce the current density when there is less or more PTh in the samples?
Response to comment 11: The existence of polythiophene layer is conducive to improving the light absorption capacity. More importantly, the combination of polythiophene and ZnO promotes the separation of photogenerated charges. When the amount of thiophene is 10 times that of ZnO, the thickness of the polythiophene layer is the most favorable for separation of photogenerated charges. A small amount of thiophene is not conducive to light absorption, and a large amount of thiophene increases the probability of recombining of photogenerated electrons and holes in the polythiophene layer. Therefore, less or more thiophene is not conducive to the improvement of photocatalytic efficiency. As shown in Figure 3e, too much or too little polythiophene will lead to the reduction of photocurrent.
Comment 12: P8, L292 and Figure 4: Did you measure the temperature profile during time? Is the illuminated quartz tube isothermal for 4 h or the photocatalytic tests in Figure 4 b-c are affected by heating? Please, can you provide a profile of the temperature during the test?
Response to comment 12: The LED lamp is equipped with a heat dissipation device, and the temperature of the reaction mixture in the quartz tube remains constant at 28 oC during the reaction process. The photos the LED lamp is shown in Figure a,b below, and the temperature change curve of the reaction mixture with time is shown in Figure c below. Due to the fact that the heat generated by LEDs is much smaller than that of xenon lamps, the temperature at each stage of the reaction using LEDs as light source remains stable at 28 oC.
Comment 13: P6, L207: In Figure 4a, in which condition the samples were compared, in more detail, can you indicate the time of reaction? Is it 4 h?
Response to comment 13: Since the content of polythiophene layer has a great impact on the photocatalytic activity of the composite catalyst, Fig. 4a is the comparison of the content of polythiophene on the photocatalytic performance. Unless otherwise specified, all photoreaction time in the manuscript is 4 h. Corresponding modifications have been made in the revised manuscript.
Comment 14: P6, L207: Can you compare the performance of your photocatalysts to those obtained in the literature for similar samples?
Response to comment 14: The comparison with published literatures on photocatalytic CO2 reduction related to ZnO or polymers are shown in the table below.
Photocatalyst |
Light Source |
Product |
Production Rate |
Ref. |
ZnO/PTh10 |
LED |
CH4 |
5.35 μmol g-1 h-1 |
This Work |
CO |
2.05 μmol g-1 h-1 |
|||
Ni-NiS/C/ZnO |
350W Xe lamp |
CH4 |
1.14 μmol g-1 h-1 |
Nanoscale, 2020, 7206-7213 |
CO |
5.86 μmol g-1 h-1 |
|||
CsPbBr3 NC/BZNW/MRGO |
300W Xe lamp |
CH4 |
0.86 μmol g-1 h-1 |
J. Mater. Chem. A, 2019, 13762-13769 |
CO |
5.86 μmol g-1 h-1 |
|||
ZnMn2O4/ZnO |
300W Xe lamp |
CH4 |
0.34 μmol g-1 h-1 |
Chem. Eng. J., 2021, 127377 |
CO |
3.2 μmol g-1 h-1 |
|||
AuNPs@SCX4+ |
300W Xe lamp |
CH4 |
0.23 μmol g-1 h-1 |
ACS Appl. Mater. Interfaces, 2022, 30796-30801 |
CO |
1.69 μmol g-1 h-1 |
|||
ZnO/graphene |
300W Xe lamp |
CH4 |
0.09 μmol g-1 h-1 |
Chem. Eng. J., 2021, 128501 |
CO |
3.38 μmol g-1 h-1 |
|||
CH3OH |
0.59 μmol g-1 h-1 |
|||
ZnO-110 |
UV-vis light |
CO |
0.76 μmol g-1 h-1 |
ACS Appl. Mater. Interfaces, 2020, 56039–56048 |
Ag-Cu2O/ZnO |
300W Xe lamp |
CO |
3.36 μmol g-1 h-1 |
Appl. Catal. B Environ., 2020, 118380 |
ZnO/NiO-30 |
300W Xe lamp |
CH3OH |
1.57 μmol g-1 h-1 |
J. CO2 Util., 2018, 548-554 |
CN-5% AP |
3W LED |
CO |
1.34 μmol h-1 |
Chin. J. Catal., 2023, 91-102 |
Comment 15: P6, L207: Did the photocatalysts produce hydrogen or other by-products? If affirmative, can you provide the yield of these secondary products?
Response to comment 15: This photocatalytic reaction only produces CH4 and CO, and does not produce any other gas or liquid phase products. This conclusion has been confirmed through gas chromatography and liquid chromatography detection.
Comment 16: P8, L294: Which is the overall efficiency of the photocatalytic process?
Response to comment 16: According to the literature (10.1038/s41560-019-0431-1), the calculation process of quantum efficiency is as follows:
Quantum efficiency of CH4:
where Y is the yield of CH4 evolution for the sample, N is Avogadro’s number, T is the irradiation time, θ is the photon flux and S is the illumination area. The following calculation example is based on the data from CO2 photoreduction with ZnO/PTh10 for 4 h: Y = 0.214 × 10−4 mol, N = 6.022 × 1023 mol−1, T = 4 h, S = 23.55 cm2; integration photons 420–800 nm, θ = 1.49 × 1017 s−1 cm−2.
For ZnO/PTh10:
the quantum efficiency of CH4: QE% = (8 × 0.214 × 10−4 × 6.022 × 1023)/(1.49 × 1017 × 4 × 3,600 × 23.55) = 0.204%;
the quantum efficiency of CO: QE% = (2 × 0.071 × 10−4 × 6.022 × 1023) / (1.49 × 1017 × 4 × 3,600 × 23.55) = 0.017 %;
The overall efficiency of ZnO/PTh10: QE% = 0.204% + 0.017 % = 0.221%.
Meanwhile, the quantum efficiencies of CH4 and CO for the pristine ZnO is calculated to be 0.047 % and 0.020 %, respectively. The overall efficiency of ZnO/PTh10: QE% = 0.047 % + 0.020 % = 0.067%.
Comment 17: P8, L292: Which is the size of the illuminated cross-section area?
Response to comment 17: The diameter of the quartz tube is 2.5 cm, and the height of the reaction mixture is 3 cm. The cross-section area is 2.5×3.14×3=23.55 cm2.
Comment 18: Lastly, I warmly recommend a careful revision of the English grammar and of the sentences in the whole manuscript.
Response to comment 18: We have rechecked the entire manuscript and made revisions to inappropriate or even incorrect sentences and words, which are marked with yellow.

Round 2
Reviewer 2 Report
I would like to thank the authors for their great efforts to answer the questions and comments of the reviewers. In my opinion, the whole research work has been improved and could be accepted in its present form for publication in Catalysts.